# Chitosan/Phenolic Compounds Scaffolds for Connective Tissue Regeneration

**DOI:** 10.3390/jfb14020069

**Published:** 2023-01-28

**Authors:** Beata Kaczmarek-Szczepańska, Izabela Polkowska, Katarzyna Paździor-Czapula, Beata Nowicka, Magdalena Gierszewska, Marta Michalska-Sionkowska, Iwona Otrocka-Domagała

**Affiliations:** 1Department of Biomaterials and Cosmetic Chemistry, Faculty of Chemistry, Nicolaus Copernicus University in Torun, Gagarin 7, 87-100 Toruń, Poland; 2Department and Clinic of Animal Surgery, Faculty of Veterinary Medicine, University of Life Sciences in Lublin, Akademicka 13, 20-950 Lublin, Poland; 3Department of Pathological Anatomy, Faculty of Veterinary Medicine, University of Warmia and Mazury, Oczapowskiego 13, 10-719 Olsztyn, Poland; 4Department of Physical Chemistry and Polymer Physical Chemistry, Faculty of Chemistry, Nicolaus Copernicus University in Torun, Gagarin 7, 87-100 Toruń, Poland; 5Department of Environmental Microbiology and Biotechnology, Faculty of Biology and Veterinary Science, Nicolaus Copernicus University in Toruń, Lwowska 1, 87-100 Torun, Poland

**Keywords:** chitosan, phenolic acids, in vivo, histological assessment

## Abstract

Chitosan-based scaffolds modified by gallic acid, ferulic acid, and tannic acid were fabricated. The aim of the experiment was to compare the compatibility of scaffolds based on chitosan with gallic acid, ferulic acid, or tannic acid using the in vivo method. For this purpose, materials were implanted into rabbits in the middle of the latissimus dorsi muscle length. A scaffold based on unmodified chitosan was implanted by the same method as a control. Moreover, the Fourier transform infrared spectroscopy-attenuated total reflectance (FTIR-ATR) spectra and scanning electron microscope (SEM) observations were made to study the interactions between chitosan and phenolic acids. Additionally, antioxidant properties and blood compatibility were investigated. The results showed that all studied materials were safe and non-toxic. However, chitosan scaffolds modified by gallic acid and tannic acid were resorbed faster and, as a result, tissues were organized faster than those modified by ferulic acid or unmodified.

## 1. Introduction

Phenolic acids are natural compounds proposed as effective cross-linkers for biopolymers [1]. They are able to form strong hydrogen bonds with functional groups present in the polymeric chain, such as hydroxyl and amine [2]. As a result, they improve the properties of the biopolymer-based materials, mainly the mechanical parameters. Additionally, the stability is improved as biopolymer-based materials without cross-linkers have a low resistance to the enzymes and may easily dissolve in aqueous conditions [3].

Chitin, the parent compound of chitosan, is a biopolymer present in many organisms, including exoskeletons of the crustaceans, mollusks, algae, insects, and the cell wall of fungi [4]. Chitosan is a natural polysaccharide and is obtained in the process of alkaline deacylation of chitin, as a result of which partial removal of acetyl groups from acetylamino groups and their replacement amino groups occurs. It consists of a deacetylated part (β-(1,4)-D-glucosamine) and an acetylated part (N-acetyl-D-glucosamine). Chitosan has antibacterial activity, along with antifungal, mucoadhesive, analgesic, and haemostatic properties [5]. It can be biodegraded into non-toxic residues and is noncytotoxic [6]. The rate of its degradation is highly related to the molecular mass of the polymer and its deacetylation degree. All these singular features make chitosan an outstanding candidate for biomedical applications. 

Chitosan-based materials cross-linked by different phenolic acids have already been studied [7]. Tannic acid classified into a polyphenolic group presents unique antiviral and antibacterial properties [8]. It has many hydroxyl groups which easily interact with functional groups of chitosan. Therefore, tannic acid was considered as a chitosan cross-linker. The results showed that the mechanical properties and surface free energy were improved, and films showed antibacterial activity against *Staphylococcus aureus* [9]. In vitro studies were also carried out, where the normal and cancer cells (MNT-1, SK-MEL-28, Saos-2, HaCaT and BMSC) were seeded on the film composed of chitosan and tannic acid. The results showed that films with the lowest tannic acid (CTS/TA 80/20) content inhibit the cell growth of MNT-1 cells; however, the lowest inhibition was observed for BMSC [10]. It was also found that a safe method for the sterilization of films enriched by phenolic acids may be using UVC light (254 nm). Moreover, it was noticed that the addition of ferulic acid to chitosan causes effective antimicrobial activity [11]. It was also reported that chitosan-based films cross-linked by tannic acid degrade in soil and compost. After 14 days of biodegradation, the chemical structure of the materials was changed, resulting from adhesion of the microorganisms. Additionally, chitosan modified by gallic acid has been studied. Obtained films showed antimicrobial activity, and the improvement of mechanical properties was observed [12]. Chitosan/gallic acid-based materials showed antioxidant activity and were noncytotoxic [13]. Moreover, scaffolds based on chitosan modified by gallic acid were studied upon their application in tissue engineering. They showed good compatibility with NIH 3T3 L1 cells and the adhesion of fibroblasts was confirmed by SEM analysis [14]. Based on the obtained results, it was assumed that the proposed films were interesting for medical application due to their nontoxicity and antibacterial activity [15]. The last stage for the primary studies of chitosan-based materials modified by selected phenolic acids includes in vivo tests.

The aim of the study is to compare the properties of scaffolds obtained from chitosan modified by gallic acid, ferulic acid, and tannic acid for the application in tissue engineering. In this study, the SEM observation and FTIR-ATR spectra were made. Additionally, antioxidant activity and blood compatibility studies were carried out. A pre-clinical in vivo animal test was performed in a tracheal defected rabbit model for 4 weeks to confirm epithelium, cartilage regeneration and biodegradability of scaffolds that are important factor to be considered [16]. Chitosan scaffolds modified by gallic acid, ferulic acid, and tannic acid are safe for medical use and are potentially suitable for cartilage tissue regeneration.

## 2. Materials and Methods

### 2.1. Materials

Chitosan (CTS; shrimp sourced, low molecular weight; Sigma-Aldrich, Poznan, Poland) was used for the studies. Gallic acid (GA; M_w_ = 170.12 g/mol), ferulic acid (FA; M_w_ = 194.19 g/mol), and tannic acid (TA; M_w_ = 1701.23 g/mol) were purchased from the Sigma-Aldrich company (Poznań, Poland).

### 2.2. Chitosan Characterization

Chitosan (low molecular weight) has been purchased from the Sigma-Aldrich company. The deacetylation degree (DD, %) of chitosan was determined by potentiometric titration, and its viscosity-average molecular weight by viscosimetery in the CH_3_COOH/CH_3_COONa solvent was as we described in detail earlier [17]. The viscosity average Mv of chitosan was equal to 375 kDa and the deacetylation degree 76.46 ± 0.22%. 

### 2.3. Preparation of Samples

Chitosan, tannic acid, gallic acid, and ferulic acid were dissolved in 0.1M acetic acid at 1% concentration separately. Chitosan solution was mixed with 10 (*w*/*w*%) addition of phenolic acid solution for 1 h. The mixtures were then placed in the 24-well sterile cell culture plates (2.5 mL/well), frozen for 24 h in −18 °C, and lyophilized (−20 °C, 100 Pa, 48 h, ALPHA 1–2 LDplus, CHRIST, Berlin, Germany). As a result, the three-dimensional dry samples (scaffolds) were obtained.

### 2.4. Scanning Electron Microscope (SEM)

The morphology of the samples was studied using a scanning electron microscope (SEM) (LEO Electron Microscopy Ltd, Thornwood, NY, USA). Samples were covered with gold before analysis. Scanning electron microscope images were made with magnification 200×.

### 2.5. Optical Microscope 

The morphology of wet scaffolds was studied using the optical microscope Motic SMZ-171 BLED (Hong Kong, China) in magnification 10×. Scaffolds were immersed in PBS for 1 h before observation. Then, they were transferred under the microscope and images were taken. 

### 2.6. Fourier Transform Infrared Spectroscopy—Attenuated Total Reflectance (FTIR-ATR)

FTIR-ATR spectra were made for each scaffold in the range 4000–600 cm^−1^ with the spectrometer Nicolet iS5 (Thermo Fisher Scientific, Waltham, MA, USA) equipped with a ZnSe crystal. Spectra were recorded with the resolution 4 cm^-1^ and 32 scans. The spectra were acquired in the absorbance mode.

### 2.7. Mechanical Properties

Mechanical properties were measured with the use of a mechanical testing machine equipped with compression jigs (EZ-Test SX TextureAnalyzer, Shimadzu, Kyoto, Japan). Scaffolds in cylindrical shapes sized 20 mm in diameter and 13 mm in height were used for the study. The samples were introduced between two discs and compressed (the starting speed of 5 mm/min up to 60% of strain) [18]. The elastic modulus (Young’s modulus, E) was calculated from the slope of the stress–strain curve in the linear region (strain from 0.20% to 0.60%). The results were recorded by Trapezium X software (version 1.4.5, Shimadzu, Kyoto, Japan).

### 2.8. Swelling Rate

Scaffolds (dry with known weight, W_dry_) were immersed in PBS for 8 hours to study the swelling behavior. After 1, 2, 3, 4, 5, 6, 7 and 8 h scaffolds were gently dried with tissue paper and weighed (W_wet_). The swelling rate was then calculated from the equation:(1)$$\mathrm{Swelling}\;\mathrm{rate}=\frac{\left(W_{\mathrm{wet}}-W_{\mathrm{dry}}\right)}{W_{\mathrm{dry}}}*100\%$$

### 2.9. Antioxidant Properties

The antioxidant properties (RSA% = radical scavenging activity) of the scaffolds were determined using the DPPH reagent (2,2-diphenyl-1-picrylhydrazyl, free radical, 95%; Alfa Aesar, Germany). Samples were placed in a 24-well plate and filled with 2 mL of a DPPH solution (250 µM solution in methyl alcohol). After 1 h, a spectrophotometric measurement was made at 517 nm (UV-1800, Shimadzu, Reinach, Switzerland). The radical scavenging activity was calculated from the formula [19]:(2)$$\mathrm{RSA}\%=\frac{{\mathrm{Abs}}_{\mathrm{DPPH}}-{\mathrm{Abs}}_{\mathrm{PB}}}{{\mathrm{Abs}}_{\mathrm{DPPH}}}*100\%$$
where Abs_DPPH_ is the absorbance of the DPPH solution without contact with the material being tested, and Abs_PB_ is the absorbance of the DPPH solution after contact with the material being tested.

### 2.10. Blood Compatibility

A total of 0.2 mL of anticoagulated (by citrate phosphate dextrose adenine addition) sheep blood was added to 10 mL of physiological saline solution containing different specimens. Positive ([OD]positive) and negative ([OD]negative) samples were prepared by adding 0.2 mL of fresh blood to water and physiological saline, respectively, and incubated at 37 °C. After 1 h, the suspension was centrifuged (1000 rpm, 10 min) and absorbance of the supernatant of each tube was measured at 540 nm with the microplate reader Multiscan FC (Thermo Fisher Scientific, Waltham, USA). Hemolysis rate was calculated using the equation (n = 3) [20]:(3)$$\mathrm{rate}\;\mathrm{of}\;\mathrm{hemolysis}\;\left[\%\right]=\frac{\left[\mathrm{OD}\right]\mathrm{specimen}\;-\left[\mathrm{OD}\right]\mathrm{negative}}{\left[\mathrm{OD}\right]\mathrm{positive}-\left[\mathrm{OD}\right]\mathrm{negative}}*100\%$$

### 2.11. In Vivo Experiment

The in vivo experiment was carried out on a group of male New Zealand rabbits weighing 2.8–3.2 kg (n = 4). The rabbits were purchased at the Experimental Medicine Center of the Medical University of Silesia in Katowice, fak. No.KCM/FPS/0041/06/20. Before the procedure, the animals’ health was checked. The animals were under constant veterinary supervision and were given a vaccine: Castomix by Pharmagal Bio against Myxomatosis (MXT) and rabbit hemorrhagic disease (RHDV). All research protocols were carried out in accordance with the ethical standards and recommendations for accommodation and care of laboratory animals covered by the European Directive 2010/63/EU on the protection of animals used for scientific purposes and the Local Ethics Committee of the University of Life Sciences in Lublin No. 104/2017, and the experiment was conducted in accordance with the provisions on animal protection. The animals were placed in the animal facility of the Experimental Medicine Center of the Medical University of Lublin. During this time, their natural habits were monitored and the temperature of each animal was measured daily. The general condition of the rabbits was very good, with no clinical signs of disease. The daily measured body temperature was within the reference range.

For 7 days after the herd was introduced to the Vivarium, their body temperature was measured and the food intake and the behavior of the animals were observed. After a week’s adaptation, the rabbits were prepared for surgery. After weighing, each individual was premedicated. On the day of surgery, each animal in the group was sedated by an intramuscular injection (Domitor-Orion Corporation, Espoo, Finland) of medetomidine (0.5 mg/kg) and Butomidor (Richter Pharma, Budapest, Hungary) butorphanol (0.2 mg/kg) depending on their weight. Then, after about 15 min, a mask was put on in order to administer inhalation anesthesia (isofluorane).

The period of anesthesia of each individual lasted about 30 min. After the rabbit was immobilized, the skin was shaved and disinfected with alcohol and iodine. The material for implantation was prepared according to the recommendations.

The skin incision was made parallel, in the intercostal area, in the middle of the latissimus dorsi muscle length, 3 cm above the dorsal line. Subcutaneous tissue and fascia were dissected in the same line and the prepared material was placed (cylindrical shape height 1 cm, diameter 1.5 cm). Two materials were implanted into the one organism (one on the left one on the right side). Experimental samples were chitosan modified by gallic acid (CTS_GA), ferulic acid (CTS_FA) and tannic acid (CTS_TA). The control was prepared by the implantation of chitosan scaffold without the addition of phenolic acid (CTS). The implantation site was closed with a mattress suture using Dexon 3-0. 

After the operation, all the rabbits could move freely in the cages without additional dressings in the area operated on. In order to minimize the risk of infection and reduce postoperative discomfort, an antibiotic and an anti-inflammatory drug (gentamicin 5 mg/kg and meloxicam 0.4 mg/kg) were administered for 5 days after the procedure.

In the postoperative period, a mild swelling was observed around the skin suture in most rabbits. After two weeks, all the operated animals were in good general condition. Three months after surgery, all rabbits were sacrificed. First, the animals were anesthetized intramuscularly and sedated by intramuscular injection of medetomidine (0.5 mg/kg) and butorphanol (0.2 mg/kg), depending on their weight.

The rabbits were then sacrificed by barbiturate injection. Tissue fragments with a margin (3cm × 3cm × 3cm) were taken from the implantation site along with the implanted material and placed in a buffered paraformaldehyde solution at pH 7.4. All the samples were placed in appropriate transporters and accurately described according to the implanted material.

### 2.12. Histological Assessment

Tissue samples were immediately fixed in 10% buffered formalin, processed routinely for histopathology using the paraffin method, cut at 5 µm and stained with Mayer’s hematoxylin and eosin. Samples were evaluated in a blind fashion by an experienced pathologist (IOD). Microphotographs were prepared using an Olympus BX43 microscope (Tokyo, Japan), equipped with an Olympus SC 180 camera (Hamburg, Germany) and cellSens software (Olympus).

### 2.13. Statistical Analysis

Statistical analysis of the data was performed using commercial software (SigmaPlot 14.0, Systat Software, San Jose, CA, USA). The Shapiro–Wilk test was used to assess the normal distribution of the data. All the results were calculated as means ± standard deviations (SD) and statistically analyzed using one-way analysis of variance (one-way ANOVA). Multiple comparisons versus the control group between means were performed using the Bonferroni t-test with the statistical significance set at *p* < 0.05.

## 3. Results

### 3.1. Scanning Electron Microscope (SEM)

All tested scaffolds had homogeneous porous structures. The biggest pores were observed for chitosan-based scaffolds with diameters more than 500 µm, as well as with diameters around 150 ± 25 µm (Figure 1). Scaffolds with phenolic acids showed a more homogeneous structure, with a pore diameter 120 ± 22 µm for chitosan/gallic acid, 112 ± 18 µm for chitosan/ferulic acid, and 115 µm ± 20 µm chitosan/caffeic acid. The addition of phenolic acids results in a decrease in the pores’ diameter. However, all pores form an interconnected network, which is highly useful for tissue engineering.

### 3.2. Optical Microscope

Dry scaffolds showed a porous structure with open pores. However, the swelling process of the scaffold results in the closure of pores for each type of scaffold (Figure 2). 

### 3.3. Fourier Transform Infrared Spectroscopy—Attenuated Total Reflectance (FTIR-ATR)

The FTIR spectra of chitosan scaffolds modified by phenolic acids are shown in Figure 3. Characteristic peaks for chitosan can be determined on each spectrum without shifts, such as the peak at 3115 cm^−1^ from NH_2_ and OH groups, at 1605 cm^−1^ from C=O and NH groups, at 1540 cm^−1^ from CN and NH groups, at 1372 cm^−1^ from CH_2_ groups, at 1058 cm^−1^ and 1032 cm^−1^ from COH and CH_2_OH. A shift of peak from COC (1205 cm^−1^ for CTS_FA, 1208 cm^−1^ for CTS_GA, 1211 cm^−1^ for CTS_FA, 1290 cm^−1^ for CTS) can be noticed that suggests the changes in the conformation of chitosan after the addition of phenolic acids [15]. The spectra of each study’s scaffolds are analogous, as phenolic acids have similar functional groups, such as carboxyl and hydroxyl ones. Moreover, comparing spectra with pure chitosan, significant changes are not observed. This suggests that covalent bonds are not formed, and the chitosan cross-linking by phenolic acids is based on the hydrogen bonds between hydrophilic groups.

### 3.4. Mechanical Properties

Mechanical parameters such as Young’s modulus (E_mod_) and maximum force (F_max_) were determined for each type of scaffold (Table 1). The highest parameters were noticed for chitosan/gallic acid scaffolds. The lowest mechanical parameters were observed for chitosan-based scaffolds without phenolic acid addition (Figure 4).

### 3.5. Swelling Rate

The swelling rate of the scaffolds based on chitosan and chitosan modified by the addition of gallic acid, ferulic acid, and tannic acid is shown in Figure 5. The swelling rate of all scaffolds was similar. The lowest swelling rate was noticed for chitosan modified by ferulic acid.

### 3.6. Antioxidant Properties

Antioxidant properties of biomaterials are useful as they can inhibit reactive oxygen species (ROS) that cause oxidative stress. The addition of phenolic acids improved the radical scavenging activity of chitosan scaffolds. Chitosan modified by gallic acid, ferulic acid, and tannic acid eliminates harmful ROS excess and inhibits molecular oxidation. The RSA values are similar and do not depend on the type of phenolic acid (Table 2).

### 3.7. Blood Compatibility

One of the most critical criteria to consider for materials in blood-material contact for biomedical applications is the rate of hemolysis. Materials are acceptable that show hemolysis lower than 5%. According to the ASTM F756-00 standard, materials with a hemolytic index between 0 and 2% are classified as non-hemolytic, while materials with 2–5% are slight hemolytic and >5% are classified as hemolytic [21]. The results of the blood compatibility measurement are listed in Table 3. The hemolysis for all tested scaffolds is below 2%. It suggests that materials are nonhemolytic.

### 3.8. Histological Assessment

In the control sample (Figure 6A), within the subcutaneous adipose tissue fragments of fibrous material (fragments of scaffold) were observed. These fragments were surrounded by a moderate infiltration of lymphocytes, macrophages, and moderately numerous multinucleated giant cells (of foreign body type) with proliferation of the connective tissue.

Figure 6B shows the tissue after implantation of the CTS_GA sample; no implant was observed. In the skin, focal proliferation of cell-poor, collagen-rich connective tissue was observed, with the subsequent atrophy of the adnexa (cutaneous fibrosis). It suggests that the scaffold was resorbed and replaced by connective tissue. Tissues undergo organization relating to the healing and repair of the tissue.

In the CTS_FA sample (Figure 6C), the partial (minimal) resorption of the implant was observed. The large implant was located within the subcutaneous tissue, surrounded by an infiltration of macrophages, single multinucleated giant cells and neutrophils, and variably numerous lymphocytes, with peripheral proliferation of the connective tissue. 

In the CTS_TA sample (Figure 6D), no implant was observed. There was a slight cutaneous fibrosis with the atrophy of adnexa, but indiscernible grossly.

## 4. Discussion

Scaffolds obtained from chitosan and with gallic acid, ferulic acid, as well as tannic acid, had a porous structure with interconnected pores. Chitosan/phenolic acids scaffolds showed a more regular and dense morphology than pure chitosan. Their structure is appropriate and allows the consideration of the material for tissue engineering purposes [22]. 

The FTIR-ATR spectra of chitosan and chitosan/phenolic acid scaffolds were studied. Peak shifts are not observed between the spectrum obtained for chitosan, chitosan/gallic acid, chitosan/ferulic acid, and chitosan/tannic acid. All components have hydrophilic structures and cross-linking of chitosan bases on the hydrogen bonds formation (Figure 7). Therefore, new functional groups are not formed, and there are no changes in the FTIR spectra. Similar results were obtained by Woranuch et al. [23] for ferulic acid-grafted chitosan. We confirmed what is also reported; that phenolic acids interact with chitosan by hydrogen bond formation [24]. 

Biomaterials with antioxidant properties are useful as they inhibit harmful ROS. Chitosan without phenolic acids has no antioxidant activity [25]. Improving the antioxidant activity of chitosan can be observed by adding small molecule compounds with antioxidant activity on chitosan molecules [26]. It has been reported that the addition of caffeic acid and ferulic acid grafted onto chitosan improves the antioxidant activity of chitosan [27,28]. It has been proven that the scavenging activity was higher for chitosan/caffeic acid material than for chitosan/ferulic acid. Additionally, Marzano et al. [29] studied the antioxidant activity of chitosan/gallic acid complexes and assessed the DPPH radical scavenging assay as much higher than free chitosan. Woranuch et al. [23] obtained ferulic-grafted chitosan that showed greater radical scavenging activity than chitosan. Chitosan functionalized with tannic acid showed improved antioxidant activity [30]. In our studies, the addition of gallic acid, ferulic acid, and tannic acid provides novel properties of chitosan-based scaffolds. The antioxidant activity of all studied scaffolds with selected phenolic acids was similar, above 90%, which provides very good radical scavenging activity. 

Hemocompatibility is an important criterion for successful biomaterial application. Erythrocytes are the most rigid cells in the blood. Erythrocytes are sensitive to hemolysis due to the shear stress [31]. Materials that have contact with blood should be nonhemolytic and not cause erythrocytes lysis. Chitosan-based hydrogels with ferulic acid and poly(vinyl alcohol) were determined as hemocompatible and having excellent biocompatibility [32]. Hydrogels based on quaternized chitosan with tannic acid showed hemocompatibility below 5% [33]. Therefore, they are considered to be safe in the wound healing treatment. Additionally, hemolysis below 2% was observed for hydrogels based on starch, carboxymethyl chitosan, and hyaluronic acid with tannic acid [34]. 

The hypodermis is composed of adipose and collagen. In deep skin defects, it takes a long period of time for re-epithelialization to be complete. Additionally, on the skin scars may be formed as a base. Traditional surgeries sutures and mechanical fixation are the most widely used methods for wound healing treatment. Wound healing is a constant process that can be defined by four phases: hemostasis, inflammation, proliferation and remodeling/maturation [35]. Interactions of biomaterial with the surrounding environment after implantation should be studied to consider its biocompatibility. It has been reported that chitosan-based materials result in the formation of a foreign body granuloma.

Scaffolds showed a high capability to swell, and it may be a reason that mild swelling was observed around the skin suture in most rabbits. In the control sample with the implanted chitosan scaffold, the regeneration of the skin was observed. However, small fragments of chitosan scaffold were also still visible, surrounded by inflammatory cells and fibrous tissue. No formation of blood vessels was noticed. Formed tissue was compared with the skin after material implantation. For the CTS_GA sample, the collagen-rich connective tissue was observed. However, the subsequent atrophy of the adnexa (cutaneous fibrosis) was noticed. Cutaneous fibrosis is the accumulation of extracellular matrix (ECM) components in the dermis, leading to compromised function and altered architecture of the dermis. The formation of fibrosis occurs as a natural process in scar formation. During fibrosis two major processes are carried out, such as synthesis and degradation of ECM, which are normally in equilibrium. It is important to shift the processes on the synthesis side during skin regeneration [36]. CTS_GA samples did not show such a trend. Hydrogels obtained from chitosan grafted with gallic acid were investigated by Sun et al. [37] as a bio-adhesive wound healing dressing. It was concluded that chitosan/gallic acid hydrogels promote wound healing. Through their favorable hemostasis, anti-inflammatory and fibroblast proliferation resulted. The implantation of gallic acid-loaded chitosan nanoparticles infused with collagen-fibroin for four days resulted in the granulation tissue formation. On day 8 of wound healing, the formation of fewer blood vessels along the established thick epithelial layer was observed. On day 12 a great amount of blood vessel formation with prominent epidermis and mature dermis regeneration was observed [38]. Ferulic acid is considered as safe with low toxicity. It possesses many physiological functions such as anti-inflammatory, antimicrobial, antithrombothic, and antidiabetic activity [39]. Ferulic acid is also considered to have an angiogenesis effect. Wounds after treatment by ferulic acid were almost completely healed after 16 days [40]. Considering our studies on CTS_FA, the material was still observed in the implantation place. Additionally, the chitosan-based scaffolds modified by ferulic acid showed the lowest swelling rate. It was not resorbed completely and was located within the subcutaneous tissue, surrounded by an infiltration of macrophages, single multinucleated giant cells and neutrophils, and variably numerous lymphocytes, with peripheral proliferation of the connective tissue. This suggests that collagen surrounding the extracellular matrix was formed even if the scaffold was not degraded [41]. For comparison, a CTS_TA scaffold was not observed. It was resorbed without any immunological reaction. Slight cutaneous fibrosis was observed, but was indiscernible grossly. Our studies provide a good conclusion about the safety of the implantation of chitosan-based hydrogels with gallic acid, ferulic acid, tannic acid, and allow the assumption that they are appropriate for skin regeneration. 

## 5. Conclusions

Chitosan scaffolds modified by gallic acid, tannic acid, and ferulic acid showed porous structures with open pores in a dry state. However, for swelled scaffolds, the closure of pores was observed. Mechanical parameters of scaffolds increased after the addition of phenolic acids, for Young’s modulus from 19.41 kPa for chitosan to 30.61 kPa for chitosan/gallic acid, 27.74 kPa for chitosan/ferulic acid, and 32.42 kPa for chitosan/tannic acid. A similar trend was observed for the maximum force—an increase from 0.75 N for chitosan to 1.28 N for chitosan/gallic acid, 0.97 N for chitosan/ferulic acid, and 1.21 N for chitosan/tannic acid. The antioxidant activity significantly increases after the addition of phenolic acids to more than 91%. All scaffolds showed a hemolysis rate below 2% and are considered safe to be used in contact with blood. Scaffolds were successfully implanted into the subcutaneous tissue of rabbits. The histological images allowed a comparison of the tissue regeneration processes occurring after scaffold implantation. It may be assumed that all scaffolds are safe for medical use, but chitosan scaffolds modified by gallic acid and tannic acid were resorbed faster and tissues are organized faster than those modified by ferulic acid. All tested materials are considered biocompatible and safe. 

## Figures and Tables

**Figure 1 jfb-14-00069-f001:**
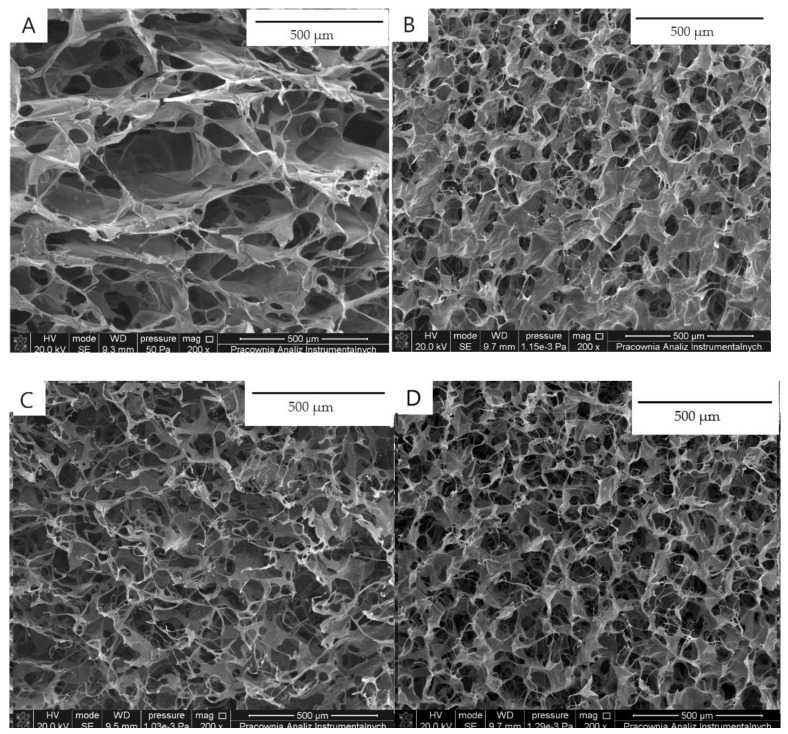
SEM images of (**A**) chitosan, (**B**) chitosan/gallic acid, (**C**) chitosan/ferulic acid, (**D**) chitosan/tannic acid.

**Figure 2 jfb-14-00069-f002:**
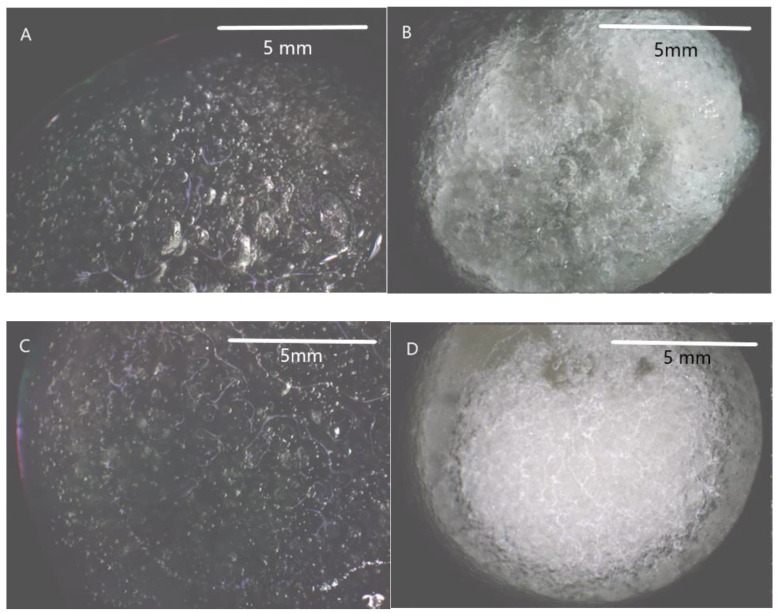
The microscope images of swollen scaffolds of (**A**) chitosan, (**B**) chitosan/gallic acid, (**C**) chitosan/ferulic acid, (**D**) chitosan/tannic acid.

**Figure 3 jfb-14-00069-f003:**
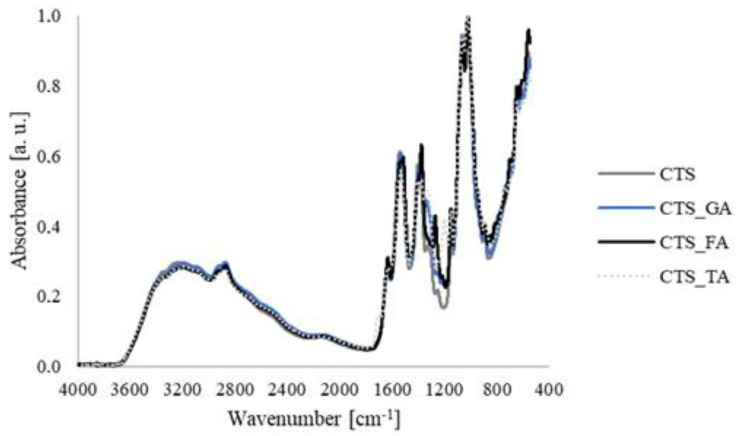
The FTIR-ATR spectra for chitosan (CTS), chitosan/gallic acid (CTS_GA), chitosan/ferulic acid (CTS_FA), and chitosan/tannic acid (CTS_TA) scaffolds.

**Figure 4 jfb-14-00069-f004:**
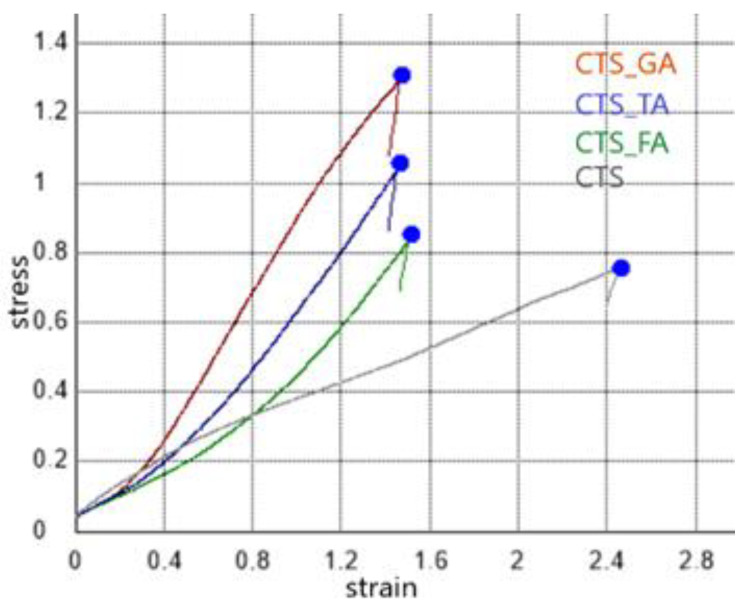
Representative stress–strain curves for chitosan (CTS), chitosan/gallic acid (CTS_GA), chitosan/tannic acid (CTS_TA), and chitosan/ferulic acid (CTS_FA).

**Figure 5 jfb-14-00069-f005:**
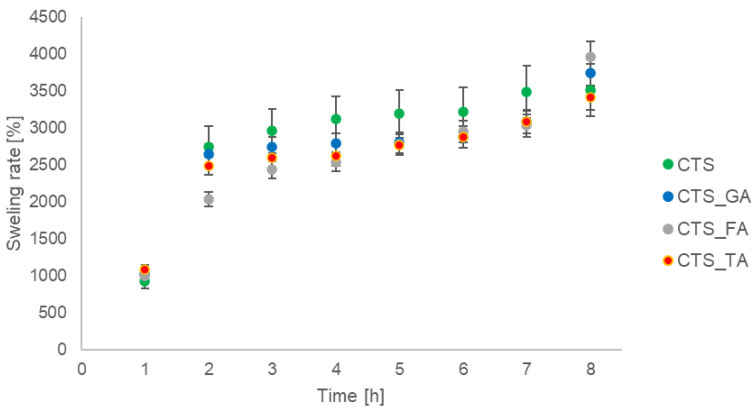
The swelling rate for chitosan (CTS), chitosan/gallic acid (CTS_GA), chitosan/tannic acid (CTS_TA), and chitosan/ferulic acid (CTS_FA).

**Figure 6 jfb-14-00069-f006:**
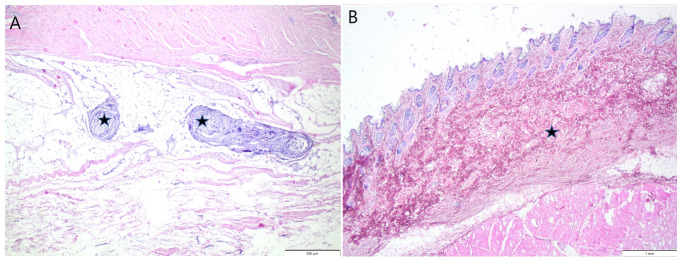
Tissue with implanted (**A**) chitosan scaffold (asterisks—fragments of scaffolds), (**B**) chitosan/gallic acid scaffold (asterisk—collagen-rich connective tissue), (**C**) chitosan/ferulic acid (asterisks—fragments of scaffold), (**D**) chitosan/tannic acid (asterisk—subsequent cutaneous fibrosis).

**Figure 7 jfb-14-00069-f007:**
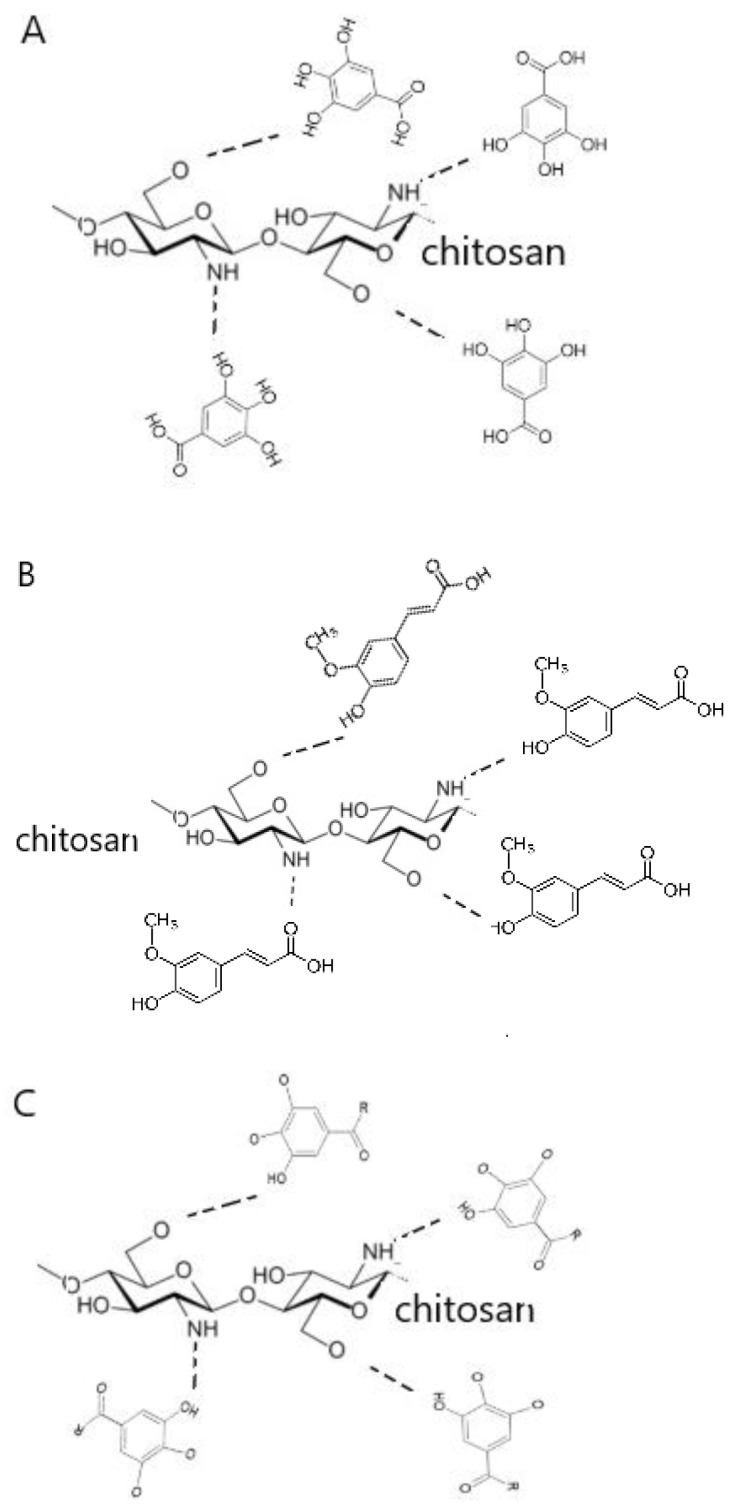
The formation of hydrogen bonds between chitosan and (**A**) gallic acid, (**B**) ferulic acid, (**C**) tannic acid.

**Table 1 jfb-14-00069-t001:** The mechanical parameters for all tested samples.

Sample	E_mod_ [kPa]	F_max_ [N]
CTS	19.41 ± 0.21	0.75 ± 0.09
CTS_GA	30.61 ± 0.17 *	1.28 ± 0.02 *
CTS_FA	27.74 ± 0.12 *	0.97 ± 0.11 *
CTS_TA	32.42 ± 0.35 *	1.21 ± 0.14 *

n = 5; * significantly different from CTS—*p* < 0.05.

**Table 2 jfb-14-00069-t002:** The radical scavenging activity (RSA) obtained for all tested samples.

Sample	RSA [%]
CTS	22.19 ± 0.15
CTS_GA	91.61 ± 0.21 *
CTS_FA	91.65 ± 0.17 *
CTS_TA	91.56 ± 0.19 *

n = 5; * significantly different from CTS—*p* < 0.05.

**Table 3 jfb-14-00069-t003:** The hemolysis rate for all tested samples (n = 5; * significantly different from CTS—*p* < 0.05).

Sample	Hemolysis Rate [%]
CTS	1.59 ± 0.21
CTS_GA	1.12 ± 0.17 *
CTS_FA	0.98 ± 0.15 *
CTS_TA	1.74 ± 0.10

## Data Availability

The data presented in this study are available on request from the corresponding author. The data are not publicly available due to an ongoing study.

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
