# Peer review of "Chitosan/Phenolic Compounds Scaffolds for Connective Tissue Regeneration"

_jfb, 2023, doi:10.3390/jfb14020069_

Round 1
Reviewer 1 Report
The research paper entitled: Chitosan/phenolic compounds scaffolds for connective tissue regeneration could be interesting for the readers of this journal. Despite that, some problems with the quality of the paper must be solved. A major revision should be carried out. The following issues I would like to comment:
1. The introduction is too short. The chitosan resources should be better mentioned in the introduction. A separation should also be made between extraction and natural resources of chitosan. The use of chitosan in the field of biomedical applications is huge. Alternative ways of using chitosan-based structures for biomedical applications should be mentioned. The following paper should be cited, as it is an interesting alternative: 10.1002/marc.201600795. Also the aim of this study is not really clear from the introduction.
2. The names of bacteria should be written in italic and In vivo too.
3. The chemical cross-linking should be shown as structural formulas, as reference 14 is not providing this.
4. All used chemical should be stated out as (purity/molecular weight, manufacturer, city, country).
5. All used devices should be stated out in the same way as it has been done in line 86.
6. Line 77: -18oC? Please fix it.
7. All equations should be numbered. For all equation also a reference is missing.
8. Figure 1: Add suitable scale bars. The images are also too dark. Please provide better ones.
9. Figure 2 is in black and white and too small. In this way it is not possible to clearly see the separate spectra line. Also the y-axis is shown in comma separations for the values. The main peaks with wavenumber must be labelled. What is what peak indicating? What is the unit of the absorbance here? At least (a. u.) should indicated with values shown. Also, add the full names of all abbreviations to the figure caption.
10. Define RSA in the table caption.
11. Figure 3 to 6 should be combined in a logical way. There is no need to stretch the manuscript. Also add in the figure caption the meaning of the stars in the micrographs.
12. The conclusion is not suitable. Add the reason of this study, present the main results in a proper way and provide an outlook for the use of the results.
13. The study has a lack of measurements, which are important for the interpretation of the results. For SEM, the samples must be dry. Provide images of the samples undried. How is the real porosity of all samples? Give an overview of the porosity and provide the pore sizes for each sample.
14. What a about hardness and flexibility of the samples? How much they are suitable to put them into the body? Please provide the hardness, Young’s modulus and stress-strain measurements.
15. For the cell settlement, some parameters are important. Provide the wettability and the surface energies (also disperse and polar component) for each sample.
Author Response
- The introduction is too short. The chitosan resources should be better mentioned in the introduction. A separation should also be made between extraction and natural resources of chitosan. The use of chitosan in the field of biomedical applications is huge. Alternative ways of using chitosan-based structures for biomedical applications should be mentioned. The following paper should be cited, as it is an interesting alternative: 10.1002/marc.201600795. Also the aim of this study is not really clear from the introduction.
Thank you very much for the comment. Introduction section is now improved. The aim of the study is specified.
- The names of bacteria should be written in italic and In vivo too.
Thank you very much for the comment. It is now corrected.
- The chemical cross-linking should be shown as structural formulas, as reference 14 is not providing this.
Thank you for the comments. It is now shown in the discussion section.
- All used chemical should be stated out as (purity/molecular weight, manufacturer, city, country).
Thank you very much for the suggestion. All details are now written in the manuscript.
- All used devices should be stated out in the same way as it has been done in line 86.
It is now corrected.
- Line 77: -18oC? Please fix it.
It is now corrected.
- All equations should be numbered. For all equation also a reference is missing.
Thank you very much for the suggestion. Equations are numbered and references are written.
- Figure 1: Add suitable scale bars. The images are also too dark. Please provide better ones.
Thank you for the comment. It is now corrected.
- Figure 2 is in black and white and too small. In this way it is not possible to clearly see the separate spectra line. Also the y-axis is shown in comma separations for the values. The main peaks with wavenumber must be labelled. What is what peak indicating? What is the unit of the absorbance here? At least (a. u.) should indicated with values shown. Also, add the full names of all abbreviations to the figure caption.
The figure 2 is now corrected.
- Define RSA in the table caption.
It is now defined.
- Figure 3 to 6 should be combined in a logical way. There is no need to stretch the manuscript. Also add in the figure caption the meaning of the stars in the micrographs.
Thank you for the comment. It is now corrected.
- The conclusion is not suitable. Add the reason of this study, present the main results in a proper way and provide an outlook for the use of the results.
Thank you for the comment. The conclusion section is now improved.
- The study has a lack of measurements, which are important for the interpretation of the results. For SEM, the samples must be dry. Provide images of the samples undried. How is the real porosity of all samples? Give an overview of the porosity and provide the pore sizes for each sample.
Thank you very much for the comment. It is now discussed and images of wet scaffolds are included.
- What a about hardness and flexibility of the samples? How much they are suitable to put them into the body? Please provide the hardness, Young’s modulus and stress-strain measurements.
Thank you for the suggestion. The mechanical parameters were determined and are now included in the paper.
- For the cell settlement, some parameters are important. Provide the wettability and the surface energies (also disperse and polar component) for each sample.
Thank you very much for the suggestion. We absolutely agree that such parameters are important. Thereby, we studied the wettability and the surface energies by contact angle measurement. However, we could not determine the contact angle of droplets placed on the surface of porous material. Please find images of the measurement below:

Reviewer 2 Report
The authors have submitted an interesting article Chitosan/phenolic compounds scaffolds for connective tissue regeneration" which deals with in vivo compatibility of polyphenol-modified chitosan platforms to boost connective tissue repair. The manuscript is well structured and reads well overall, although it will need some revisions. I suggest this article be published after a serious major revision.
*** General comments:
ü The abstract is clear and concise and comprises all cornerstones including a brief/general introduction to the topic, a non-technical summary of the significant findings, and their implications.
ü The introduction is too short. It needs t be developed.
ü The experimental design is logical, however, there are still some comments to be covered and some concerns to be addressed.
ü The conclusions are logical but short.
*** Suggested revisions:
1- First of all, I strongly recommend the authors provide a simple, high-quality, and informative “Graphical abstract” which can present the whole concept of your study at a glance. I would like to recommend authors design a “Graphical Abstract” for this study to better show the whole story in a simple and informative manner. In this regard, you can illustrate a simple sketch of the big picture and add elements like SEM images and histology.
2- Please carefully revise the manuscript to remove grammatical errors and vague sentences. Some of the sentences are unnecessary which makes it difficult and boring for the readers to follow them. Please double-check the whole manuscript and revise all.
3- In the introduction part to better present the fundamentals of this field, please read and add valuable information from the following key paper (synthetic polyphenols developed from phenolic building blocks like polydopamine should be introduced)
Page 1, line 31: Mussel-inspired biomaterials: From chemistry to clinic - https://doi.org/10.1002/btm2.10385
4- The conclusion section is short and insufficient. Please develop it based on the obtained data.
Author Response
1- First of all, I strongly recommend the authors provide a simple, high-quality, and informative “Graphical abstract” which can present the whole concept of your study at a glance. I would like to recommend authors design a “Graphical Abstract” for this study to better show the whole story in a simple and informative manner. In this regard, you can illustrate a simple sketch of the big picture and add elements like SEM images and histology.
Thank you for the suggestion. Graphical abstract is now added.
2- Please carefully revise the manuscript to remove grammatical errors and vague sentences. Some of the sentences are unnecessary which makes it difficult and boring for the readers to follow them. Please double-check the whole manuscript and revise all.
Thank you very much for the comment. Manuscript was doble checked and corrected.
3- In the introduction part to better present the fundamentals of this field, please read and add valuable information from the following key paper (synthetic polyphenols developed from phenolic building blocks like polydopamine should be introduced) Page 1, line 31: Mussel-inspired biomaterials: From chemistry to clinic - https://doi.org/10.1002/btm2.10385
Thank you for the suggestion. The introduction section is now improved.
4- The conclusion section is short and insufficient. Please develop it based on the obtained data.
Thank you for the comment. The conclusion section is now improved.

Reviewer 3 Report
The authors submitted an article entitle "Chitosan/phenolic compounds scaffolds for connective tissue regeneration ". Here are my comments:
1- The chemical interaction/molecular structure between chitosan and gallic acid, ferulic acid, and tannic acid should be illustrated.
2- There are several reports regarding chitosan-gallic acid scaffolds, therefore the novelty and necessity of this work is under question and should be clarified in the abstract.
3- Be consistent in using abbreviations. For example, "SEM" has been used in abstract and again in headline 3.1 "Scanning electron microscope (SEM)".
4- Add quantification of pore size analyzed by SEM. Rearrange the figure and give some space between images.
5- Add more details to the conclusion part.
Author Response
Reviewer #3:
1- The chemical interaction/molecular structure between chitosan and gallic acid, ferulic acid, and tannic acid should be illustrated.
Thank you very much for the comment. It is now shown in the paper.
2- There are several reports regarding chitosan-gallic acid scaffolds, therefore the novelty and necessity of this work is under question and should be clarified in the abstract.
It is now specified in the abstract.
Thank you very much for the comment. Chitosan/gallic acid materials are now discussed in the introduction section.
3- Be consistent in using abbreviations. For example, "SEM" has been used in abstract and again in headline 3.1 "Scanning electron microscope (SEM)".
Thank you for this suggestion. It is now corrected.
4- Add quantification of pore size analyzed by SEM. Rearrange the figure and give some space between images.
Thank you very much for the comment. The pore size is now discussed and figures are rearranged.
5- Add more details to the conclusion part.
Thank you for the comment. The conclusion section is now improved.

Round 2
Reviewer 1 Report
The authors addressed the most of my comments adequately and they improved their manuscript significantly, but I have some points why should be addressed:
1. The molecular weight of the chemicals you added in chapter 2.1 are written with comma. Please use the notification style with a point, as this style is common and requested.
2. Often space signs are missing for units and equal signs. For example: n=5, 48h or Mw=17... Please check the whole manuscript to correct this issue.
3. The new graphical abstract is strange and not really attractive to me. A good abstract should following this rules: 1. what material, 2. what modification and most important measuring method (if suitable), 3. example how this could be used in the best way. And to show already some gained results in a graphical abstract is mostly also not so good. Try to improve the graphical abstract.
4. I saw the image of the try to measure the contact angles. After which time the image was taken? Which droplet settlement time was used in this experiment? After 1 minute maybe the droplet is away. Which liquid was used? If only water, what about other common liquids for wettability test (diiodmethane, formamide, ...) ? Do all samples showed the same behaviour?
5. How much is swelling of each samples? Please provide the swelling rate of each sample. According to figure 2, the swelling behaviour should be different. Maybe the swelling of the samples is the explanation for the sentence in line 186-187 or was it an inflammation process?
Author Response
Dear All,
on behalf of myself and co-authors, I am enclosing the manuscript jfb-2119005 entitled “Chitosan/phenolic compounds scaffolds for connective tissue regeneration” that we believe should be of strong interest to the general readership of the Journal of Functional Biomaterials.
We would like to note that in addition to addressing all reviewer’s valuable remarks, the authors placed additional editorial corrections including references to improve the quality of the manuscript. Below are our point-by-point responses to reviewer’s comments and colored in yellow.
Reviewer #1:
- The molecular weight of the chemicals you added in chapter 2.1 are written with comma. Please use the notification style with a point, as this style is common and requested.
Thank you very much. It is now corrected.
- Often space signs are missing for units and equal signs. For example: n=5, 48h or Mw=17... Please check the whole manuscript to correct this issue.
Thank you for the comment. It is now corrected.
- The new graphical abstract is strange and not really attractive to me. A good abstract should following this rules: 1. what material, 2. what modification and most important measuring method (if suitable), 3. example how this could be used in the best way. And to show already some gained results in a graphical abstract is mostly also not so good. Try to improve the graphical abstract.
Thank you very much for the comment. The graphical abstract is now changed.
- I saw the image of the try to measure the contact angles. After which time the image was taken? Which droplet settlement time was used in this experiment? After 1 minute maybe the droplet is away. Which liquid was used? If only water, what about other common liquids for wettability test (diiodmethane, formamide, ...) ? Do all samples showed the same behaviour?
The droplet is away immediately after the settlement. It was also observed for other liquids (glycerine, diiodmethane, formamide) as materials are porous. Thereby, we were not able to carry out the wettability studies. All samples showed the same behaviour.
- How much is swelling of each samples? Please provide the swelling rate of each sample. According to figure 2, the swelling behaviour should be different. Maybe the swelling of the samples is the explanation for the sentence in line 186-187 or was it an inflammation process?
Thank you very much for the comment. The swelling rate studies are now included in the paper and the results are discussed.
Reviewer 2 Report
The manuscript is well-amended and it is ready for publication. I have no further comments.
Author Response
Thank you very much.
Reviewer 3 Report
The authors improved the quality of the manuscript. I have no further comments.
Author Response
Thank you very much.